# Nasopharyngeal carriage of *Streptococcus pneumoniae* in Latin America and the Caribbean: A systematic review and meta-analysis

**Martín Brizuela**[1]*, **María Carolina Palermo**[2], **Tomás Alconada**[2], **María Macarena Sandoval**[2], **Eugenia Ramirez Wierzbicki**[2], **Joaquín Cantos**[2], **Paula Gagetti**[3], **Agustín Ciapponi**[2,4], **Ariel Bardach**[2,4], **Silvina Ruvinsky**[5,6]

1 Unidad de Pediatría, Hospital General de Agudos Vélez Sarsfield, Buenos Aires, Argentina, 2 Instituto de Efectividad Clínica y Sanitaria (IECS-CONICET), Buenos Aires, Argentina, 3 Servicio Antimicrobianos, Laboratorio Nacional de Referencia (LNR), Instituto Nacional de Enfermedades Infecciosas (INEI)- ANLIS "Dr. Carlos G. Malbrán", Buenos Aires, Argentina, 4 Centro de Investigaciones Epidemiológicas y Salud Pública (CIESP-IECS) CONICET, Buenos Aires, Argentina, 5 Coordinación de Investigación. Hospital de Pediatría Dr. Juan P. Garrahan, Buenos Aires, Argentina, 6 Departamento de Evaluación de Tecnologías Sanitarias y Economía de la Salud. Instituto de Efectividad Clínica y Sanitaria, Buenos Aires, Argentina

* martin.brizuela1984@gmail.com

**Data Availability Statement:** All relevant data are within the manuscript and its Supporting information files.

## Abstract

### Background

*Streptococcus pneumoniae* is a leading cause of morbidity and mortality globally, causing bacteremic pneumonia, meningitis, sepsis, and other invasive pneumococcal diseases. Evidence supports nasopharyngeal pneumococcal carriage as a reservoir for transmission and precursor of pneumococcal disease.

### Objectives

To estimate the pneumococcal nasopharyngeal burden in all age groups in Latin America and the Caribbean (LAC) before, during, and after the introduction of pneumococcal vaccine conjugate (PVC).

### Methods

Systematic literature review of international, regional, and country-published and unpublished data, together with reports including data from serotype distribution in nasopharyngeal carriage in children and adults from LAC countries following Cochrane methods. The protocol was registered in PROSPERO database (ID: CRD42023392097).

### Results

We included 54 studies with data on nasopharyngeal pneumococcal carriage and serotypes from 31,803 patients. In children under five years old, carriage was found in 41% and in adults over 65, it was 26%. During the study period, children under five showed a

**Funding:** All authors have received funding by Pfizer Global Medical Grants (GMG) 76436251. The funders had no role in study design, data collection and analysis, decision to publish, or preparation of the manuscript.

**Competing interests:** The authors have declared that no competing interests exist.

colonization proportion of 34% with PCV10 serotypes and 45% with PCV13 serotypes. When we analyze the carriage prevalence of PCV serotypes in all age groups between 1995 and 2019, serotypes included in PCV10 and those included in PCV13, both showed a decreasing trend along analysis by lustrum.

## Conclusion

The data presented in this study highlights the need to establish national surveillance programs to monitor pneumococcal nasopharyngeal carriage to monitor serotype prevalence and replacement before and after including new pneumococcal vaccines in the region. In addition, to analyze differences in the prevalence of serotypes between countries, emphasize the importance of approaches to local realities to reduce IPD effectively.

## Introduction

*Streptococcus pneumoniae* is a significant cause of morbidity and mortality worldwide producing different types of infections such as bacteremic pneumonia, meningitis, sepsis and other invasive pneumococcal diseases (IPD). Evidence supports nasopharyngeal pneumococcal carriage as a reservoir for transmission and precursor of pneumococcal disease [1, 2].

Complex relationships between *S. pneumoniae* and the human host were described. The main reservoir of *S. pneumoniae* is the upper airway, allowing transmission. Host factors are essential for developing pneumococcal disease, mainly acute otitis media, pneumonia, bacteremia, and meningitis. The colonization, transmission, and invasion are related to the capacity to evade immune and inflammatory host responses [3, 4].

As with pneumococcal disease, the epidemiology of pneumococcal carriage varies by age; the highest rate among children < 5 years was reported [3].

Pneumococcal conjugate vaccines (PCVs) are safe and effective and have been used in high-income countries for many years. They have now been widely adopted worldwide. Vaccination with PCV provides herd immunity, reducing the transmission to households and others in the community, and preventing pneumococcal infection in the unvaccinated population [5].

Following the introduction of pneumococcal conjugate vaccines (PCVs) into national immunization programs (NIPs), IPD and nasopharyngeal carriage was reduced in upper-middle and high-income countries [6]. This reduction has been observed primarily among vaccinated children and all ages, including the elderly [7].

In LAC countries, PCVs have been introduced in their NIPs since 2007 with different strategies ("3+1" and "2+1") [8]. An impact in reducing pneumococcal vaccine-serotypes carriage and increasing non-vaccine serotypes was observed [9]. Despite evidence of almost complete replacement of serotypes in many settings, non-vaccine (NVT) serotypes have replaced vaccine-type (VT) serotypes in colonization and invasive diseases [10, 11].

Data on the epidemiology of pneumococcal carriage in children and adults after the introduction of PCV10 and PCV13 in the region are scarce. Understanding the dynamics of nasopharyngeal colonization in developing regions is crucial to predicting the public health implications of routine PCVs' use in these settings [12].

This systematic review and meta-analysis aimed to add epidemiological information about pneumococcal nasopharyngeal carriage distribution since PVCs introduction in all age groups in LAC.

## Material and methods

We conducted this systematic literature review of international, regional, and country-published and unpublished data, together with reports of routinely recorded data such as registries and MoH, including data from serotype distribution in nasopharyngeal carriage in children and adults from LAC countries.

The Cochrane methods and the 2020 Preferred Reporting Items for Systematic Reviews and Meta-Analyses (PRISMA) were followed [13] and the established guidelines for Meta-analysis Of Observational Studies in Epidemiology (MOOSE). The protocol was registered in PROSPERO database (ID: CRD42023392097) [14, 15].

### Eligibility criteria

We included cohort studies, case-control, cross-sectional studies, epidemiological surveillance reports, hospital-based surveillance studies, case series, control arms of randomized/quasi-randomized controlled trials, controlled before and after (CBA) and uncontrolled before and after (UBA) studies, interrupted time series (ITS), and controlled ITS (CITS), assessing serotyping of nasopharyngeal carriage in LAC. No language restriction was imposed. Studies with at least 20 nasopharyngeal swabs of patients of all ages, published or reported since January 2000, were included. Every Latin American and Caribbean country represented the geographical scope.

Systematic reviews and meta-analyses were considered a source of primary studies. In those cases where data or data subsets were reported in more than one publication, the one with the larger sample size was selected.

### Data sources and search strategy

We searched the primary literature, international and regional databases, generic and academic internet searches, and meta-search engines. We searched records from the following databases up to 27 December 2022: MEDLINE (PubMed), EMBASE (Elsevier interface), LILACS/ Scielo, EconLIT (EBSCO Interface), Global Health (OVID), CINAHL (EBSCO Interface), and Web of Science. We also consult websites from the leading regional medical societies, experts, and associations related to the topic. An annotated search strategy for grey literature was included to retrieve information from relevant sources like regional MoH, PAHO, and reports from hospitals. This strategy comprises two blocks plus search limits (from 2000 onwards to cover ten years of pre-vaccination programs) and a geographical string block for LAC. The search had no language restrictions and was limited to humans. The reference lists of the articles were hand-searched for additional information. After consulting the principal investigator, we selected only the publication with the largest sample size. Highly cited authors were contacted to obtain missing or extra information.

**Selection of studies, data extraction, and assessment of the risk of bias.** Two researchers reviewed each identified record by title and abstract and obtained the full texts of potentially eligible studies. The full-text articles were reviewed for compliance with the inclusion criteria. The selection process was conducted via COVIDENCE [16, 17]. This form was piloted on ten papers to refine the process. If studies had multiple publications, we collated the various reports of the same study under a single study ID with multiple references. Data from eligible studies were extracted by two reviewers using a data extraction form. Two reviewers independently performed all steps. Any disagreements were resolved by discussion with the whole team.

We independently assessed the risk of bias in observational cohort, case-control, cross-sectional, and case-series studies with the National Institute of Health (NIH) Quality Assessment

Tool. After answering the different signaling questions "Yes", "No", "Cannot determine", "Not applicable", or "Not reported", the raters classified the study quality as "Good", "Fair", or "Poor". For consistency with the other designs, we use the classifications "Low", "Unclear" or "High risk" of bias [18]. When the criteria are rated "uncertain," we obtained more information from the study authors. Pairs of independent reviewers assessed the risk of bias. Discrepancies were solved by consensus of the whole team. For the assessment of cohort studies and cross-sectional studies, the tool comprises 14 items, while nine items apply to the case series studies. For RCTs and quasi-RCTs, we evaluated the following domains: sequence generation, allocation concealment, blinding of participants and personnel, blinding of outcome assessors, incomplete outcome data, selective outcome reporting, and other possible threats to validity [19].

### Data synthesis and analysis

Data were collected from observational, comparative, and non-comparative studies, including registries. We conducted a paired meta-analysis for outcomes for which studies were deemed comparable. To estimate the prevalence of pneumococcal serotypes, we conducted a proportional meta-analysis using R software version 4.2.2 [20, 21]. We applied an arc-sine transformation to stabilize the variance of proportions (Freeman-Tukey variant of the arc-sine square root of transformed proportions method) [22]. The pooled proportion was calculated as the back-transformation of the weighted mean of the transformed proportions, using inverse arc-sine variance weights for the fixed and random effects model. We applied DerSimonian-Laird weights for the random effects model where heterogeneity between studies was found. We calculated the $I^2$ statistic as a measure of the proportion of the overall variations in the proportion that was attributable to between-study heterogeneity. An I2 of 60–70% was considered significant heterogeneity, and below 30% was considered low heterogeneity [23].

We conducted subgroup analyses classifying the studies by five-year calendar period, country, and age group (0–5 years, 6–64 years, 65 or more years old). We included data on different *S. pneumoniae* serotypes obtained from nasopharyngeal swabs.

### Results

We retrieved 8,600 records from seven different databases. Of these registries, 4,533 references remained for the screening phase after duplicates were removed, and 414 were eligible for full-text review.

Finally, we included 54 studies with data on nasopharyngeal pneumococcal carriage and serotypes from 31,803 patients. Fig 1 and S1 Table in S1 File. The search strategy is available in S1 Annex in S1 File, and the list of studies excluded during the extraction process in S2 Table in S1 File.

The characteristics of the 54 included studies are described in Table 1. We identified 43 studies that included participants below 18 years (n = 27,491), three with an adult population (n = 739), seven studies that included both pediatric and adults (n = 3,500), and one study did not report the age of participants (n = 73).

The studies were carried out in 17 countries; the most represented countries were Brazil (n = 22 with 14,234 participants), Venezuela (n = 7 with 2,453 participants) and Mexico (n = 6 with 4,340 participants). The inclusion period of participants was from 1990 to 2018.

In 47 studies, data provided was from outpatients, in one from inpatients, and in six studies it was not reported. Nineteen studies included asymptomatic and healthy participants and three included high-risk patients. Comorbidities were reported in 12 studies; respiratory chronic diseases, malnutrition, cardiac diseases, immunodeficiency, and cancer were the most frequent.

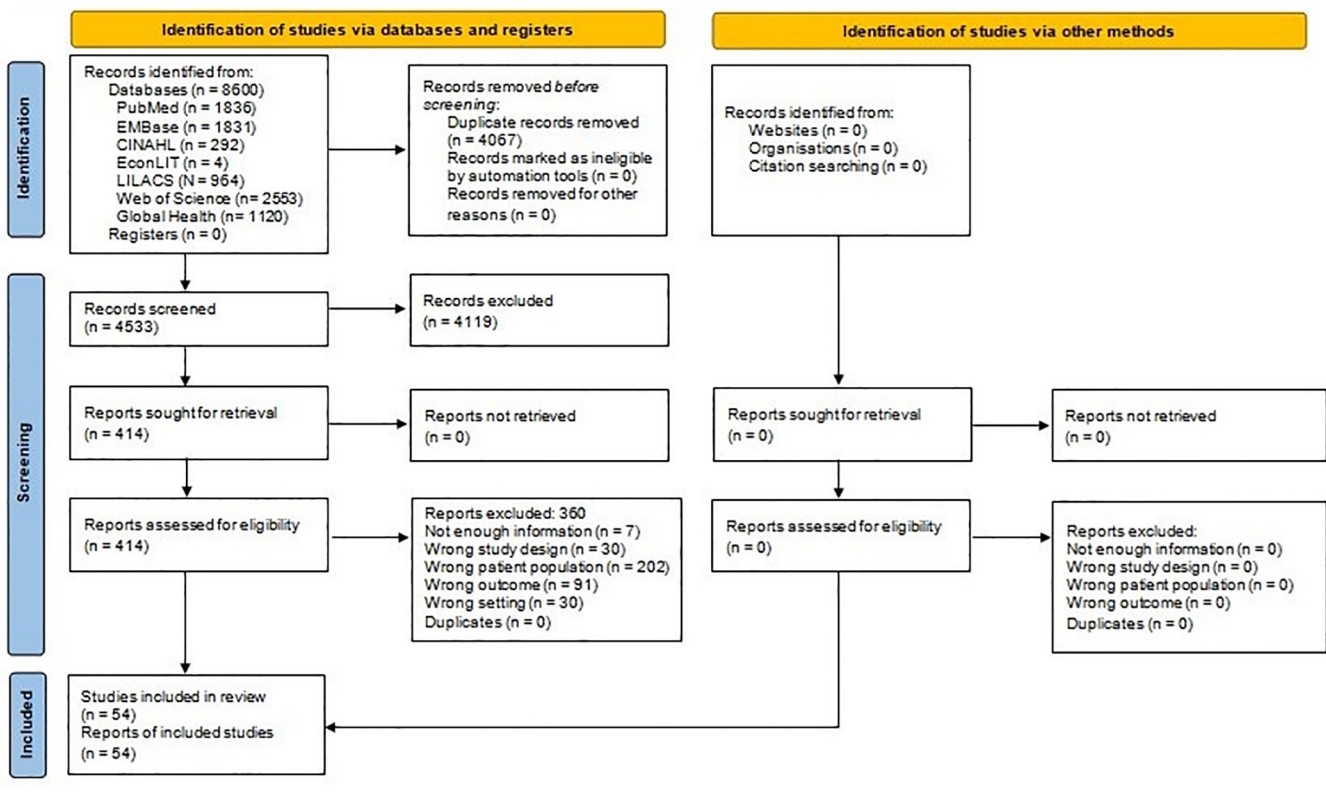

**Fig 1. Study flowchart.**

All participants came from non-probability sampling. Immunization status was reported in 37 studies: 21 included not vaccinated participants, eight studies with completely vaccinated patients, eight both (complete and incomplete vaccination), and not reported in 17 studies. Immunization scheme was: PCV7 in 5 studies, PCV10 in 3 studies, and PCV13 in 8.

Regarding the methodological design, there were 46 cross-sectional studies, including nine epidemiological surveillance studies, followed by 4 cohort studies, 3 case- series, and one case-control study.

## Meta-analysis of pneumococcal nasopharyngeal carriage

The global carriage proportion for all age groups was 38.3% (33% to 44%, CI 95%) from 1995 to 2019. In children under five years old, carriage was found in 41% (35% to 48%, CI 95%), and in adults over 65, 26% (19 to 35%, CI 95%) Figs 2 and 3. Brazil, Mexico, and Venezuela were the most representative countries with data about the nasopharyngeal carriage, including five or more studies. In Brazil, a pooled percentage of 37% (29–45%, CI 95%) was estimated, in Mexico, 34% (22–49%, CI 95%), and in Venezuela, 36% (27–47%, CI 95%). Fig 4.

When we analyzed the most representative age group to evaluate carriage, preschool children under five years old, we found a stable proportion (33–44%) among all study periods (1995–2019). In a subgroup analysis of children between 0 and 2 years old, a global nasopharyngeal carriage ratio of 40% (32–49%, CI 95%) was found. Fig 5. During the study period, children under five showed a colonization proportion of 34% (27–41%, CI 95%) with PCV10 serotypes and 45% (36–54%, CI 95%) with PCV13 serotypes. Fig 6. Only one study reported

**Table 1. Characteristics of included studies (n = 54).**

| Author and year of publication | Country | Study start date dd/mm/yyyy | Study ending date dd/mm/yyyy | Study design | Age range | Sample size |
|---|---|---|---|---|---|---|
| Wolf 2000* [24] | Brazil | 15/03/1998 | 15/12/1998 | Case control | <5y | 435 |
| Cullotta 2002* [25] | Peru | 06/06/2000 | 21/08/2000 | Case series | <5y | 302 |
| Gómez-Barreto 2002* [26] | Mexico | 01/09/1997 | 30/09/1999 | Cross sectional | NR | 73 |
| Rey 2002* [27] | Brazil | 01/01/1998 | 31/12/1998 | Cross sectional | <5y | 911 |
| Allen 2003* [28] | Jamaica | 01/06/1999 | 30/06/1999 | Cross sectional | <18y | 276 |
| Lucarevschi 2003* [29] | Brazil | 29/06/1998 | 15/12/1998 | Cross sectional | <5y | 987 |
| Ochoa 2005* [30] | Peru | 01/09/1996 | 31/12/2003 | Cross sectional/Surveillance | <2y | 666 |
| Solorzano Santos 2005* [31] | Mexico | 01/02/2002 | 31/01/2003 | Cross sectional | <5y | 573 |
| Cardozo 2006* [32] | Brazil | 01/11/2002 | 31/07/2003 | Cross sectional | <18y | 1,013 |
| Laval 2006* [33] | Brazil | 01/05/2000 | 31/08/2001 | Cross sectional/Surveillance | <5y | 648 |
| Quintero 2006* [34] | Venezuela | 01/02/2000 | 31/07/2000 | Cross sectional | <5y | 125 |
| Alturraza Hernaez 2007* [35] | Chile | 20/08/1998 | 30/10/1998 | Cross sectional | >65y | 118 |
| Berezin 2007* [36] | Brazil | 01/06/1997 | 31/05/2001 | Case series | <5y | 520 |
| Espinosa-de los Monteros 2007* [37] | Mexico | 01/09/2002 | 31/12/2002 | Cross sectional | <5y | 2,777 |
| Nicoletti 2007* [38] | Brazil | 01/03/2000 | 31/12/2001 | Cross sectional | ≥18y | 385 |
| Rivera Olivero 2007* [39] | Venezuela | 01/04/2004 | 31/01/2005 | Cross sectional/Surveillance | <5y | 356 |
| Neves Reis 2008* [40] | Brazil | 01/07/2000 | 31/05/2001 | Cross sectional | All ages | 262 |
| Reyna 2008* [41] | Mexico | 01/01/2006 | 31/12/2006 | Cross sectional | <5y | 498 |
| Bello González 2010* [42] | Venezuela | 01/05/2008 | 31/05/2008 | Cross sectional | All ages | 148 |
| Espinosa-de los Monteros 2010* [43] | Mexico | 01/02/2002 | 28/02/2004 | Prospective cohort | <2y | 183 |
| Franco 2010 [44] | Brazil | 01/08/2005 | 31/12/2005 | Cross sectional | <5y | 1,192 |
| Sisco 2010** [45] | Venezuela | NR | NR | Cross sectional/Surveillance | All ages | 488 |
| Inverarity 2011* [46] | Bolivia | 01/05/2007 | 30/06/2007 | Cross sectional | <18y | 601 |
| Quintero 2011* [47] | Venezuela | 01/01/2007 | 28/02/2007 | Cross sectional | <5y | 250 |
| Rivera Olivero 2011* [48] | Venezuela | 01/12/2006 | 31/01/2008 | Cross sectional/Surveillance | <5y | 1,004 |
| Lamaro Cardoso 2012* [49] | Brazil | 01/07/2008 | 30/09/2008 | Cross sectional | All ages | 224 |
| Lopes 2012* [50] | Brazil | 01/05/2005 | 31/01/2006 | Cross sectional | <1y | 139 |
| Mercado 2012* [51] | Peru | 01/01/2007 | 31/12/2009 | Cross sectional/Surveillance | <2y | 2,123 |
| Méndez 2013* [52] | Panama | 25/11/2008 | 19/12/2008 | Cross sectional | <5y | 397 |
| Neves 2013* [53] | Brazil | 01/03/2010 | 30/06/2010 | Cross sectional | <5y | 242 |
| Parra 2013* [54] | Colombia | 01/01/2005 | 31/12/2011 | Cross sectional | <2y | 443 |
| Andrade 2014* [55] | Brazil | 01/12/2010 | 28/02/2011 | Cross sectional | <2y | 1,287 |
| Grijalva 2014* [56] | Peru | 01/05/2009 | 30/09/2011 | Cross sectional | <5y | 309 |
| Laranjeira 2014*** [57] | Brazil | 01/01/2011 | 31/12/2011 | Cross sectional | <5y | 291 |
| Rivera Olivero 2014* [58] | Venezuela | 01/01/2007 | 31/12/2009 | Cross sectional/Surveillance | <5y | 82 |
| Menezes 2015* [59] | Brazil | 01/01/2008 | 31/01/2009 | Cross sectional | <5y | 203 |
| Brandileone 2016* [60] | Brazil | 22/03/2010 | 06/06/2013 | Cross sectional | <2y | 901 |
| Melgar 2016* [61] | Guatemala | 01/11/2012 | 31/01/2013 | Case series | <5y | 500 |
| Gentile 2017** [62] | Argentina | 01/06/2015 | 30/09/2015 | Cross sectional | <5y | 359 |
| Neves 2017* [63] | Brazil | 20/09/2014 | 05/12/2014 | Cross sectional | <5y | 522 |

*(Continued)*

**Table 1.** (Continued)

| Author and year of publication | Country | Study start date dd/mm/yyyy | Study ending date dd/mm/yyyy | Study design | Age range | Sample size |
|---|---|---|---|---|---|---|
| **Toledo Romani 2017**\* [64] | Cuba | 01/10/2013 | 30/11/2015 | Cross sectional | <5y | 2,115 |
| **Fernández 2018**\* [65] | Uruguay | 01/01/2002 | 31/12/2015 | Cross sectional/Surveillance | <2y | 831 |
| **Nelson 2018**\* [66] | Peru | 01/05/2009 | 30/09/2011 | Prospective cohort | <5y | 1,015 |
| **Brandileone 2019**\* [67] | Brazil | 16/08/2017 | 19/08/2017 | Cross sectional | <2y | 531 |
| **Cassiolato 2019** [68] | Brazil | 01/01/2005 | 31/12/2017 | Cross sectional | All ages | 1,432 |
| **Chávez Amaro 2019**\* [69] | Cuba | 01/05/2016 | 30/06/2017 | Non-comparative cohort | <5y | 555 |
| **Espinosa-de los Monteros-Pérez 2019**\* [70] | Mexico | 01/12/2009 | 30/06/2010 | Cross sectional | >50y | 236 |
| **Pinto 2019**\* [71] | Brazil | 01/01/1990 | 31/12/2014 | Cross sectional | All ages | 783 |
| **Kumar 2020**\*\* [72] | Barbados | 01/01/2015 | 31/12/2016 | Cross sectional | <5y | 216 |
| **Dunn 2021**\* [73] | Dominican Republic | 01/11/2016 | 31/07/2017 | Prospective cohort | <5y | 125 |
| **Kumar 2021**\* [74] | Barbados | 01/01/2016 | 31/12/2017 | Cross sectional | <10y | 198 |
| **Regalado 2021**\* [75] | Ecuador | 01/02/2018 | 30/04/2018 | Cross sectional | All ages | 163 |
| **Watkins 2021**\* [76] | Haiti | 01/09/2015 | 31/01/2016 | Cross sectional | <5y | 685 |
| **Silva 2022**\* [77] | Brazil | 01/02/2013 | 31/12/2013 | Cross sectional/Surveillance | <5y | 660 |

NR: Not Reported

\*Full text;

\*\*Abstract/Poster;

\*\*\*Thesis

pneumococcal nasopharyngeal carriage in the adult population more than 65 years old, and the proportion of serotypes was 51.5% (35% to 68% CI 95%) and 57.6% (41% to 73% CI 95%) for PCV10 and PCV13 respectively.

Serotypes included in PCV10 and PCV13 vaccines among nasopharyngeal carriage were analyzed by age (0- 5y, 6- 64y, and more than 65y), lustrum, and country. See S1-S8 Figs in S1 File.

We analyze the carriage prevalence of PCV serotypes in all age groups between 1995 and 2019. Serotypes included in PCV10 were 34% (28% to 40%, CI 95%), and those included in PCV13 serotypes were 45% (38% to 55%, CI 95%). Both showed a decreasing trend along analysis by five-year period. S3 Table in S1 File.

### Risk of bias in included studies

Among 46 cross-sectional and four cohorts, 31 studies (62%) were evaluated with a moderate risk of bias; 14 (28%) with a high risk, and 5 (10%) with a low risk of bias. The domains that presented the most significant inconvenience in their evaluation were those related to the sample selection, the calculation of sample size, and the measurements of the results according to different exposures. Likewise, other drawbacks in the evaluation were associated with the validation domains of the results regarding the blinding of the observers and the adjustment for possible confounding variables. Of note, other domains could not be assessed due to the nature of the studies, such as repeated measurements and follow-up in cross-sectional studies S4 Table in S1 File. The only three case series studies had a moderate risk of bias with significant inconvenience in the case definition and statistical results management. The studies did not report if the cases

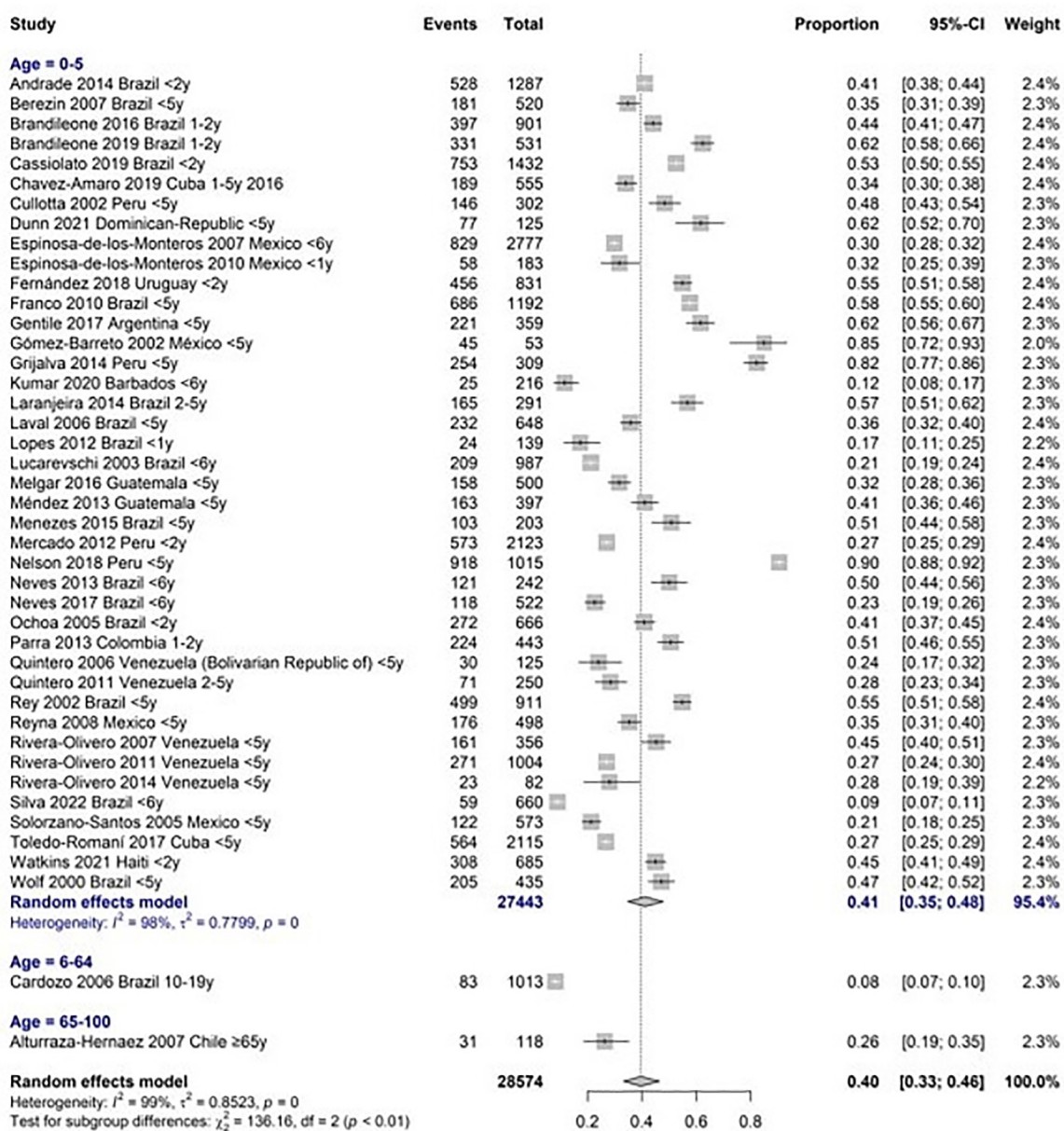

**Fig 2. Meta- analysis of nasopharyngeal carriage ratio by age group.**

were included consecutively S5 Table in S1 File. Finally, one case-control study was classified with a moderate risk of bias. The study showed the most inconvenience in the statistical management of data, sample size calculation, and blindness of observations S6 Table in S1 File.

## Discussion

*Streptococcus pneumoniae* nasopharyngeal carriage could be a precondition for pneumococcal disease and community transmission [78]. There are approximately one hundred serotypes, several producing invasive pneumococcal disease (IPD). The prevalence of nasopharyngeal

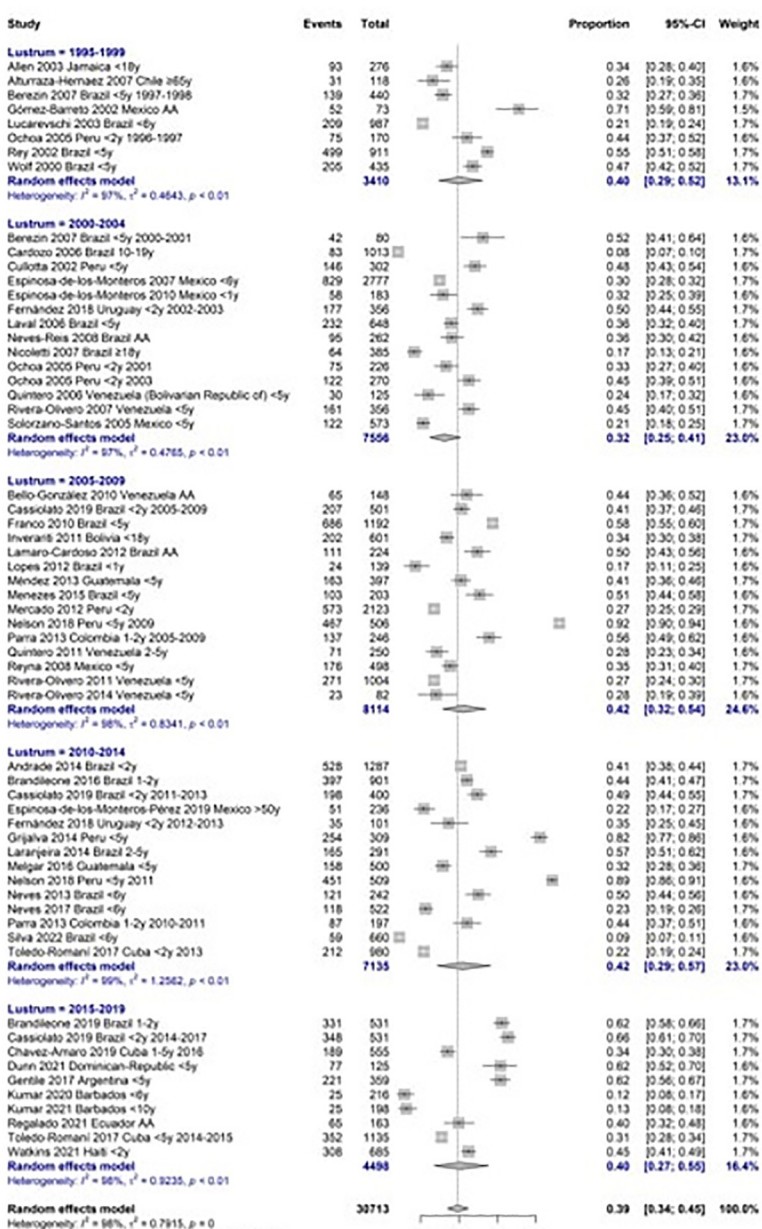

**Fig 3. Meta-analysis of nasopharyngeal carriage ratio by lustrum.**

carriage reported varies between 20–40% in healthy children and ranges from 5 to 10% among adults less than 65 years old, similar to our finding [4, 79]. While several studies showed colonization data among young children, fewer studies were performed in the elderly population. Almeida S, et al. [80] found a low global nasopharyngeal carriage (2.3%) in the elderly population living in Portugal. In our study, the data about adults older than 65 in the LAC region was found to have a higher prevalence rate of nasopharyngeal colonization, probably due to only one study being included. Pneumococcal carriage is highest during the first five years of life, peaking in the first two years in most developed countries [3].

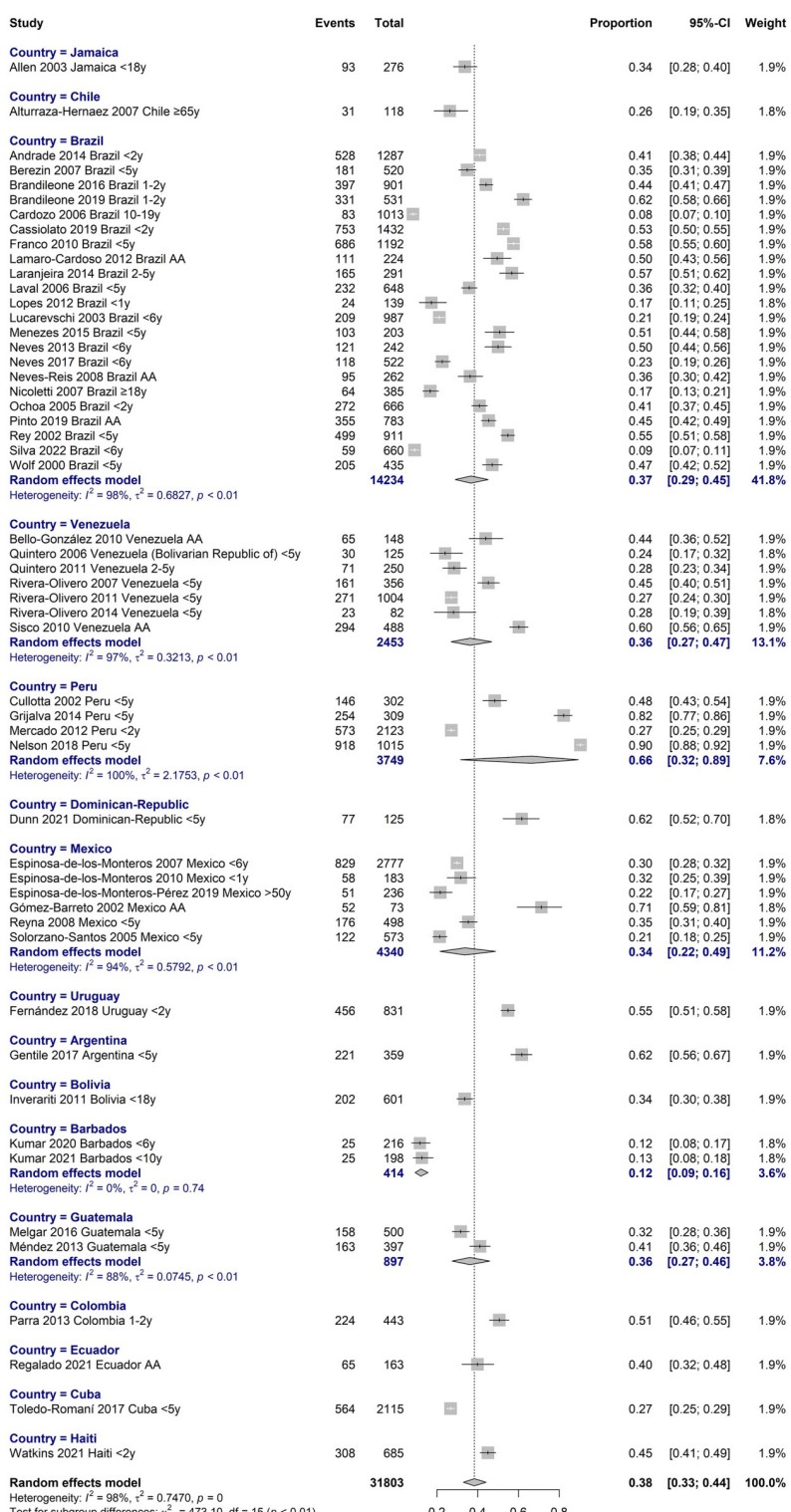

**Fig 4. Meta-analysis of *S. pneumoniae* carriage proportion by country.**

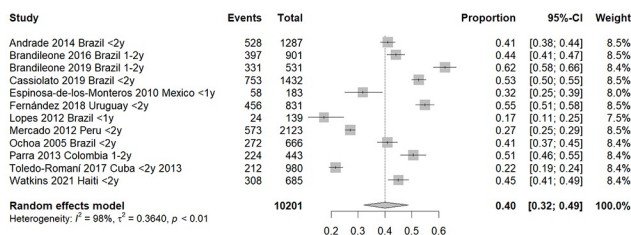

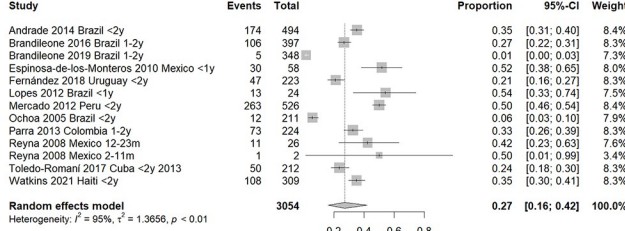

Nasopharyngeal carriage ratio (all serotypes)

Nasopharyngeal carriage ratio (PCV10 serotypes)

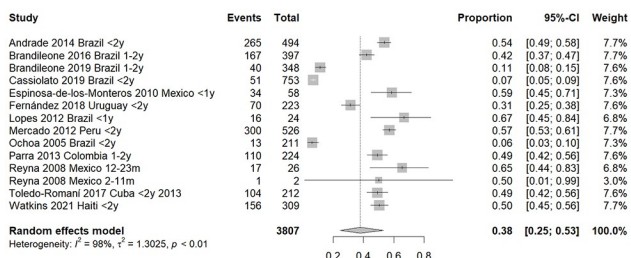

Nasopharyngeal carriage ratio (PCV13 serotypes)

**Fig 5. Meta- analysis nasopharyngeal carriage ratio among children <2 years old.**

Regev-Yochay 2004 [81] found that the rate of *S. pneumoniae* carriage was higher among children than adults in the same community. Hadinegoro 2016 [82] examined healthy children in Indonesia and identified several prevalent serotypes of *S. pneumoniae*. León 2019 [83] focused on indigenous communities in Ecuador and highlighted the need for studies on respiratory pathogen colonization in high-risk groups. Gámez 2020 [84] investigated nasopharyngeal colonization in children in southwestern Colombia and found a high frequency of *S. pneumoniae* carriage, including strains resistant to multiple antimicrobial agents.

As was described in developed and undeveloped countries, an inverse relationship between age and carriage exists [85, 86]. In line with those findings, we observed that patients under five have the highest proportion of carriage.

Disparities in the prevalence of carriage between different countries, ranging from relatively low prevalence in countries such as Barbados, Chile, and Cuba to high prevalence in Peru, Argentina, and the Dominican Republic, call for further exploration of underlying factors, such as socioeconomic conditions, health care infrastructure, and vaccination policies.

Regarding the serotype distribution in the nasopharyngeal carriage, the analysis considered the serotypes included in the PCV10 and PCV13 vaccines in the region. Previous studies have shown that the introduction of PCV10 and PCV13 in the NIP led to a decrease in pneumococcal carriage, transmission, and circulation, resulting in a reduction of IPD [87–92]. Introducing pneumococcal conjugated vaccines (PCV) in national immunization programs has increased the nasopharyngeal colonization by serotypes not included in vaccines, with a reduction of vaccine serotypes by replacement [9]. When we analyze the distribution of serotypes in pneumococcal carriage, serotypes included in PCV10 and PCV13 decreased during the last lustrum (2015–2019), probably related to serotype replacement. However, non-PCV10/

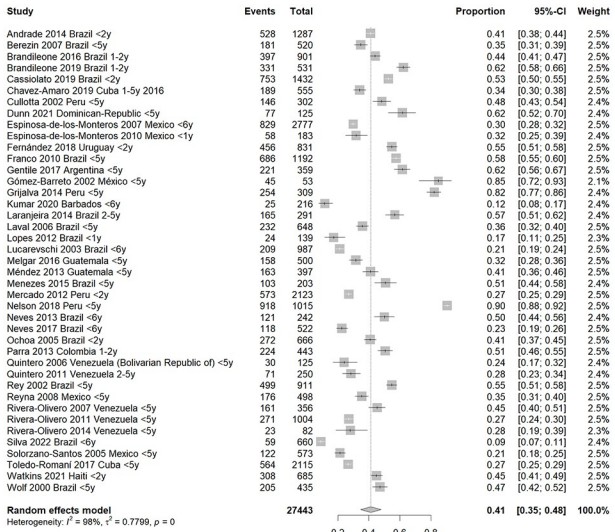

Nasopharyngeal carriage ratio (all serotypes)

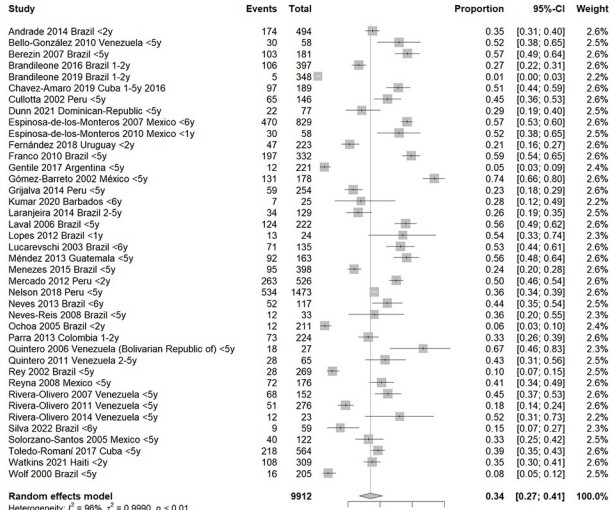

Nasopharyngeal carriage ratio (PCV10 serotypes)

Nasopharyngeal carriage ratio (PCV13 serotypes)

**Fig 6. Meta- analysis nasopharyngeal carriage ratio among children <5 years old.**

PCV13 serotypes in nasopharyngeal carriage predominate in all age groups in the region, similar to those reported by Rybak A et al. [93] The effect of vaccination on carriage offers herd protection mediated through a reduction in nasopharyngeal carriage of vaccine serotypes and pneumococcal transmission in the community.

Our study has some limitations; the heterogeneity and risk of bias of study settings and the absence of active nor passive pneumococcal nasopharyngeal carriage surveillance programs in countries in our region. Data about underlying diseases, environmental factors, or pneumococcal vaccination status of studies included were unavailable. In assessing the risk of bias, sample selection, size, and outcome measurement were the most critical aspects. Case series were mostly low or medium risk, with selection and statistical methods concerns.

The present systematic review, which includes 54 studies and covers data from 31,803 patients, offers a novel, detailed, and exhaustive panorama of the nasopharyngeal carriage of *S. pneumoniae* in the Latin American region. This extensive database provides valuable information on the epidemiology and prevalence of this pathogen, providing data on its impact on public health and possible implications for vaccination strategies. Our study contributes a baseline that could be useful to monitor the impact of new pneumococcal vaccine introduction on nasopharyngeal carriage in LAC.

In conclusion, the data presented in this study highlights the need to establish national surveillance programs to monitor pneumococcal nasopharyngeal carriage to monitor serotype prevalence and replacement before and after including new pneumococcal vaccines in the region. In addition, to analyze differences in the prevalence of serotypes between countries, emphasize the importance of approaches to local realities to reduce IPD effectively.

## Supporting information

**S1 File.**
(DOCX)

## Acknowledgments

The authors thank Mr.Daniel Comande, the Institute for Clinical Effectiveness and Health Policy librarian, for contributing to the bibliographic searches.

## Author Contributions

**Conceptualization:** Agustín Ciapponi, Ariel Bardach, Silvina Ruvinsky.

**Data curation:** Martín Brizuela, Tomás Alconada, María Macarena Sandoval, Eugenia Ramirez Wierzbicki, Joaquín Cantos, Paula Gagetti, Agustín Ciapponi, Ariel Bardach, Silvina Ruvinsky.

**Formal analysis:** Martín Brizuela, Tomás Alconada, Paula Gagetti, Agustín Ciapponi, Ariel Bardach, Silvina Ruvinsky.

**Funding acquisition:** Agustín Ciapponi, Ariel Bardach, Silvina Ruvinsky.

**Investigation:** María Carolina Palermo, María Macarena Sandoval, Agustín Ciapponi, Ariel Bardach, Silvina Ruvinsky.

**Methodology:** Ariel Bardach, Silvina Ruvinsky.

**Project administration:** Ariel Bardach.

**Resources:** Agustín Ciapponi.

**Software:** Joaquín Cantos.

**Supervision:** Paula Gagetti, Agustín Ciapponi, Ariel Bardach, Silvina Ruvinsky.

**Validation:** Martín Brizuela, María Carolina Palermo, Tomás Alconada, María Macarena Sandoval, Eugenia Ramirez Wierzbicki, Joaquín Cantos, Paula Gagetti, Agustín Ciapponi, Ariel Bardach, Silvina Ruvinsky.

**Visualization:** Martín Brizuela, María Carolina Palermo, Tomás Alconada, María Macarena Sandoval, Eugenia Ramirez Wierzbicki, Joaquín Cantos, Paula Gagetti, Agustín Ciapponi, Ariel Bardach, Silvina Ruvinsky.

**Writing – original draft:** Martín Brizuela, María Carolina Palermo, Tomás Alconada, María Macarena Sandoval, Eugenia Ramirez Wierzbicki, Joaquín Cantos, Paula Gagetti, Agustín Ciapponi, Ariel Bardach, Silvina Ruvinsky.

**Writing – review & editing:** Martín Brizuela, María Carolina Palermo, Tomás Alconada, María Macarena Sandoval, Eugenia Ramirez Wierzbicki, Joaquín Cantos, Paula Gagetti, Agustín Ciapponi, Ariel Bardach, Silvina Ruvinsky.

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
