## [Decision Letter · Decision Letter 0]

12 Dec 2023

PONE-D-23-35902Nasopharyngeal carriage of Streptococcus pneumoniae in Latin America and the Caribbean: a systematic review and meta-analysisPLOS ONE

Dear Dr. Brizuela,

Thank you for submitting your manuscript to PLOS ONE. After careful consideration, we feel that it has merit but does not fully meet PLOS ONE’s publication criteria as it currently stands. Therefore, we invite you to submit a revised version of the manuscript that addresses the points raised during the review process.

1. Review aspects of form, especially in tables

2. Table 1: It is best to sort the studies by year of publication, from 2001 to 2022.

Please submit your revised manuscript by Jan 26 2024 11:59PM . If you will need more time than this to complete your revisions, please reply to this message or contact the journal office at December 20th. Si necesitará más tiempo que este para completar sus revisiones, responda a este mensaje o comuníquese con la oficina de la revista en  plosone@plos.org. Please include the following items when submitting your revised manuscript:

We look forward to receiving your revised manuscript.

Kind regards,

Oriana Rivera-Lozada de Bonilla

Academic Editor

PLOS ONE

https://pubmed.ncbi.nlm.nih.gov/37012114/

In your revision ensure you cite all your sources (including your own works), and quote or rephrase any duplicated text outside the methods section. Further consideration is dependent on these concerns being addressed.

“All authors have received funding by Pfizer Global Medical Grants (GMG) 76436251.”

6. Please include a separate caption for each figure in your manuscript.

Reviewers' comments:

Reviewer's Responses to Questions

**Comments to the Author**

1. Is the manuscript technically sound, and do the data support the conclusions?

Reviewer #1: Yes

Reviewer #2: Yes

Reviewer #3: Yes

2. Has the statistical analysis been performed appropriately and rigorously? 

Reviewer #1: Yes

Reviewer #2: N/A

Reviewer #3: Yes

3. Have the authors made all data underlying the findings in their manuscript fully available?

Reviewer #1: Yes

Reviewer #2: Yes

Reviewer #3: Yes

4. Is the manuscript presented in an intelligible fashion and written in standard English?

Reviewer #1: Yes

Reviewer #2: Yes

Reviewer #3: Yes

5. Review Comments to the Author

Reviewer #1: Dear authors,

Thank you for this interesting review article. there are some comments which is not affect the quality of the search:

1. line 59: change Colonization to colonization.

2. line 107: change 2,022 to 2022.

3. line 161, 196: delete the word see (quotation brackets are sufficient).

4. Table 1: It's better if you sort the studies by year of publication, from 2001 to 2022.

5. line 199, 203, 205: fig. 4, 5, 6 change to figure 4, figure 5, figure 6.

Reviewer #2: The review is well designed and I hope it can be repeated with real clinical application. Not only as a review. I hope it could make equal time zone for all investigated areas. Also I wish there is a good evidence for vaccine efficacy.

Reviewer #3: The author outlined detailed information in each section of the manuscript. The data with figures are available in the manuscript. The supporting figures are available and explained very clearly. The article is clear and unambiguous.

6. PLOS authors have the option to publish the peer review history of their article (what does this mean?). If published, this will include your full peer review and any attached files.

Reviewer #1: **Yes: **Awatif Al-Judaibi

Reviewer #2: **Yes: **Dr Michael Nazmy Agban professor of Microbiology and Immunology faculty of medicine assiut university Egypt. Head of department of molecular biology at molecular biology institute

Reviewer #3: **Yes: **VenkataVinayKumar Bandarupalli

---

## [Author Response · Author response to Decision Letter 0]

8 Jan 2024

Dear editors and reviewers, thank you very much for your valuable contributions and suggestions. We have followed all suggestions and corrections to improve the quality of our research.

Below we detail the response to each of the points and have highlighted the modifications in the manuscript. Again, thank you very much.

Reviewer Request:

Table 1: It is best to order the studies by year of publication, from 2001 to 2022.

Answer:

All included studies were sorted by date.

Order:

We note that you have a minor occurrence of overlapping text with the following previous publications, which needs to be addressed: https://pubmed.ncbi.nlm.nih.gov/37012114/

Reply:

We have rephrased some material phrases and methods to improve quality and avoid overlaps.

Request financial disclosure: “All authors have received funding from Pfizer Global Medical Grants (GMG) 76436251.”

Reply:

We have added a modified feature of the funder disclosure in our cover letter and in the manuscript.

Request: We note that you have stated that you will provide repository information for your data upon acceptance. If your manuscript is accepted for publication, we will retain it until you provide the relevant accession numbers or DOIs needed to access your data. If you want to make changes to your data availability statement, please describe these changes in your cover letter and we will update your data availability statement to reflect the information you provide.

Response: We have changed all information available in the manuscript and supporting information

Request: Please include a separate title for each figure in your manuscript.

Answer: We have included a separate title for each figure in the manuscript.

Application: Please review your reference list to ensure it is complete and correct.

Answer: All references have been reviewed and are correct.

Request: Please upload a new copy of Figures 1, 2, 3, 4, 5 and 6 as the details are unclear.

Answer: The new files in figures 1 to 6 were uploaded with the best possible quality.

These specific comments and suggestions from reviewer one were modified in the manuscript and highlighted:

1. line 59: change Colonization to colonization. Response: we added The colonization....

2. line 107: change 2,022 to 2022. Response: it was changed 

3. line 161, 196: delete the word see (quotation brackets are sufficient). Response: deleted

4. Table 1: It's better if you sort the studies by year of publication, from 2001 to 2022. Response: it was sorted in a cronological order 

5. line 199, 203, 205: fig. 4, 5, 6 change to figure 4, figure 5, figure 6. Response: it was changed

---

## [Editor Report · Decision Letter 1]

12 Jan 2024

Portación nasofaríngea de Streptococcus pneumoniae en América Latina y el Caribe: una revisión sistemática y metanálisis 

PONE-D-23-35902R1

Estimado Dr. Martín Brizuela ,

Nos complace informarle que su manuscrito ha sido considerado científicamente adecuado para su publicación y será aceptado formalmente para su publicación una vez que cumpla con todos los requisitos técnicos pendientes.

Dentro de una semana, recibirá un correo electrónico detallando las modificaciones requeridas. Una vez que se hayan abordado, recibirá una carta de aceptación formal y se programará la publicación de su manuscrito.

Una factura de pago le llegará poco después de la aceptación formal. Para garantizar un proceso eficiente, inicie sesión en Editorial Manager en http://www.editorialmanager.com/pone/, haga clic en el enlace "Actualizar mi información" en la parte superior de la página y verifique que su información de usuario esté actualizada. hasta la fecha. Si tiene alguna pregunta relacionada con la facturación, comuníquese directamente con nuestro departamento de facturación de autores en Authorbilling@plos.org.

Si su institución o instituciones tienen una oficina de prensa, notifíqueles sobre su próximo artículo para ayudar a maximizar su impacto. Si van a preparar materiales de prensa, informe a nuestro equipo de prensa lo antes posible, a más tardar 48 horas después de recibir la aceptación formal. Su manuscrito permanecerá bajo estricto embargo de prensa hasta las 2 pm, hora del Este, en la fecha de publicación. Para obtener más información, comuníquese con onepress@plos.org.

Atentamente,

Oriana Rivera-Lozada de Bonilla 

Editora Académica 

PLOS ONE

---

## [Editor Report · Acceptance letter]

5 Feb 2024

PONE-D-23-35902R1 

PLOS ONE

Dear Dr. Brizuela, 

I'm pleased to inform you that your manuscript has been deemed suitable for publication in PLOS ONE. Congratulations! Your manuscript is now being handed over to our production team.

Kind regards, 

on behalf of

Dr. Oriana Rivera-Lozada de Bonilla 

Academic Editor

PLOS ONE